# The Efficacy and Safety of Neoadjuvant Immunotherapy in Patients with Non-Small Cell Lung Cancer

**DOI:** 10.3390/cancers16010156

**Published:** 2023-12-28

**Authors:** Deniz Can Guven, Taha Koray Sahin, Saadettin Kilickap

**Affiliations:** 1Medical Oncology Clinic, Health Sciences University, Elazig City Hospital, 23280 Elazig, Turkey; 2Internal Medicine Clinic, Sultanhanı Hospital, 68000 Aksaray, Turkey; koraysahin@hacettepe.edu.tr; 3Department of Medical Oncology, Istinye University Faculty of Medicine, 34010 Istanbul, Turkey; saadettin.kilickap@istinye.edu.tr

**Keywords:** non-small cell lung cancer (NSCLC), efficacy and safety, immune checkpoint inhibitors (ICIs), neoadjuvant chemoimmunotherapy, neoadjuvant immunotherapy

## Abstract

**Simple Summary:**

The benefit of neoadjuvant chemoimmunotherapy was compared with chemotherapy for localized NSCLC in several trials. We conducted a systematic review and meta-analysis to evaluate the benefit of adding immunotherapy to neoadjuvant chemotherapy in patients with localized NSCLC. We observed that neoadjuvant immunotherapy significantly improves EFS, OS, and pCR rates with a slight increase in high-grade toxicities. The subgroup analyses demonstrated a consistent benefit with neoadjuvant immunotherapy independent of age, ECOG status, smoking status, and stage, while the benefit of neoadjuvant immunotherapy was lower in patients with PD-L1-negative tumors than patients with moderate or high levels of PD-L1 expression. Further research is needed to define the optimal duration of immunotherapy after surgery and the overall survival benefit of neoadjuvant immunotherapy with longer follow-ups.

**Abstract:**

Background: After the success of immunotherapy in the treatment of advanced non-small cell lung cancer (NSCLC), the benefit of neoadjuvant chemoimmunotherapy was compared with chemotherapy for localized NSCLC in several trials. However, the available studies had variable study designs, and study cohorts had limited follow-up times. Therefore, we conducted a systematic review and meta-analysis to evaluate the benefit of adding immunotherapy to neoadjuvant chemotherapy in patients with localized NSCLC. Methods: We conducted a systematic review using Pubmed, Web of Science, and Scopus databases for studies published until 5 December 2023. This protocol was registered in the PROSPERO database (Registration Number: CRD42023466337). We performed the meta-analyses with the generic inverse-variance method with a fixed effects model. Results: Overall, 7 studies encompassing 2993 patients were included in the analyses. The use of neoadjuvant chemoimmunotherapy was associated with a 41% reduction in the risk of progression or death compared to neoadjuvant chemotherapy (HR: 0.59, 95% CI: 0.52–0.66, *p* < 0.0001) and a lower risk of death (HR: 0.67, 95% CI: 0.55–0.82, *p* < 0.0001). The neoadjuvant chemoimmunotherapy improved pCR rates compared to chemotherapy (21.8% vs. 3.8%, OR: 7.04, 95% CI: 5.23–9.47, *p* < 0.0001), while high-grade adverse events were higher with neoadjuvant chemoimmunotherapy (OR: 1.18, 95% CI: 1.02–1.36, *p* = 0.0300). Conclusions: The available evidence demonstrates a statistically significant and clinically meaningful event-free survival benefit and possibly an overall survival benefit with neoadjuvant chemoimmunotherapy with a slight increase in high-grade toxicities.

## 1. Introduction

Non-small cell lung cancer is the leading cause of cancer-related deaths due to high incidence and presentation in the advanced stage in over half of the patients and abysmal 5-year survival rates in patients with advanced-stage disease despite advances in precision medicine and immuno-oncology [1,2,3]. Furthermore, the disease recurs in over 70% and 40% of the patients with stage III and stage II disease, respectively [4]. Although adjuvant chemotherapy improved patient outcomes, the absolute 5-year overall survival benefit was only around 5% in the pivotal meta-analysis of the available clinical trials [5]. Neoadjuvant chemotherapy is suggested as a potential way to improve outcomes in localized-stage NSCLC [6]. However, low rates of pathological complete response (pCR) and a similar survival benefit to the adjuvant setting has limited its generalization in clinical practice [7]. These figures indicate a need for novel approaches for patients with localized-stage NSCLC.

The immune checkpoint inhibitors transformed the treatment algorithms in patients with advanced-stage NSCLC, either alone or in combination with chemotherapy [8,9,10]. After the benefit of ICIs in treating advanced-stage NSCLC, there is an interest in adding immunotherapy to the neoadjuvant treatment of patients with localized-stage NSCLC [11,12]. The interest in the use of ICIs in the adjuvant setting was amplified after the encouraging early results from single-arm studies like NADIM and LC3M3, demonstrating around 15–20% pCR and up to 40% major pathological response rates (MPR) and a lack of disease progression in patients with pCR after immunotherapy in the neoadjuvant setting [13,14].

After the promise of neoadjuvant immunotherapy in single-arm early-phase clinical trials, several randomized trials comparing neoadjuvant chemoimmunotherapy with chemotherapy were published [15,16,17,18,19,20,21]. However, the study designs (neoadjuvant only or neoadjuvant plus adjuvant), the patient cohorts (stages II–III or only stage III), as well as the magnitude of benefits were heterogeneous across the available studies. Additionally, the biomarkers of benefits and the subgroup analyses were not fully discussed in most studies. Therefore, we conducted a systematic review and meta-analysis to evaluate the benefit of adding immunotherapy to neoadjuvant chemotherapy in patients with localized NSCLC and conducted subgroup analyses according to clinically relevant parameters, including tumor stage, histology, smoking status, tumor PD-L1 expression, and chemotherapy partner.

## 2. Materials and Methods

### 2.1. Literature Search

This meta-analysis was registered at the PROSPERO database (No. CRD42023466337). The systematic review was conducted according to the Preferred Reporting Items for Systematic Reviews and Meta-Analysis (PRISMA) guidelines [22]. We used the PubMed, Web of Science (WOS), and Scopus databases to systematically filter the published studies until 5 December 2023 for this systemic review. The search terms and keywords were “neoadjuvant” OR “perioperative” OR “preoperative” AND “non-small cell lung cancer” OR “NSCLC” OR “non-small cell lung carcinoma” AND “immunotherapy” OR “immune checkpoint inhibitor” OR “nivolumab” OR “ipilimumab” OR “sintilimab” OR “durvalumab” OR “atezolizumab” OR “pembrolizumab” OR “avelumab” OR “camrelizumab” OR “tislelizumab” OR “tremelimumab” OR “toripalimab”. In addition, a search from the ESMO and ASCO websites was conducted for 2018–2023 to detect recent studies without full text available. The inclusion criteria were as follows: (1) randomized clinical trial comparing the efficacy of chemoimmunotherapy compared to chemotherapy in the neoadjuvant setting for the treatment of localized-stage NSCLC; (2) studies published in English; (3) available hazard ratio for event-free survival (EFS), disease-free survival (DFS) or overall survival (OS). The exclusion criteria of the study were as follows: (1) reviews, perspectives, observational studies, case reports, and study protocols; (2) single-arm clinical trials; (3) the use of ICI in both arms; (4) experimental drug other than immunotherapy in the combination arm; (5) repetitive publications of one study; (6) studies in other languages than in English. We did not exclude studies without an available full text to avoid missing recent clinical trial data. 

### 2.2. Data Extraction

Two authors independently extracted the following data from the available studies (DCG, TKS) following the PRISMA 2020 guidelines: Lead author names, year of the publication, immunotherapy agent, clinical trial phase and study design (neoadjuvant only, and neoadjuvant plus adjuvant), the total number of patients and number of patients in each arm (chemoimmunotherapy and chemotherapy), surgical resection rate, R0 resection rate, median follow-up (months), hazard ratios (HR) with 95% confidence intervals (CI) for EFS and OS, and the HR for EFS for the subgroup analyses. Due to a preference for using EFS to define progression, recurrence, or death events in the most available studies, we used the EFS as the primary measure and used the PFS or DFS data in the EFS analyses considering all three terms (EFS, PFS, and DFS) were used to define the same events across studies. Data extraction and bias assessment were independently performed by two authors (DCG and TKS), and any discrepancies were resolved through discussion with the senior author (SK). Additionally, the individual study qualities and risk of bias were evaluated independently by two authors (DCG and TKS) using the Risk of Bias Tool, Version 2 (Appendix A).

### 2.3. Meta-Analyses

The meta-analysis was performed using the generic inverse-variance method with a fixed-effects model, considering the low degree of heterogeneity across all analyses. The primary objective was to compare the EFS between the patients treated with neoadjuvant chemoimmunotherapy versus chemotherapy. In addition to comparing the EFS with chemoimmunotherapy and chemotherapy, subgroup analyses for EFS according to PD-L1 expression, smoking status, age, disease stage, and ECOG performance status were conducted. The secondary objectives were to compare the OS, pathological complete response rate (pCR), R0 resection rate, and all-grade and high-grade (grade 3 or higher) adverse events between patients treated with neoadjuvant chemoimmunotherapy versus chemotherapy. The principal summary measure were the HRs with 95% two-sided CIs for EFS and OS, and odds ratio (OR) and 95% two-sided CIs for adverse events, pCR, and R0 resection rates. All analyses were conducted using the Review Manager software, version 5.4 (The Nordic Cochrane Center, The Cochrane Collaboration, Copenhagen, Denmark). The heterogeneity within each subgroup was assessed using Higgins I-square statistics. A *p*-value below 0.05 was considered statistically significant.

## 3. Results

### 3.1. Study Selection and Baseline Study Characteristics

Our systematic search retrieved a total of 11,833 records. After removing the duplications (*n* = 10,485), we removed 1153 records after the title and abstract search. After evaluating the full texts of the remaining 195 articles, 190 records were excluded for the following reasons and a total of 6 studies were retrieved from the systematic search. One additional study and an additional survival update were identified from the search of ASCO and ESMO congresses, making a total of seven studies included in the analyses. The flowchart for study selection is demonstrated in Figure 1.

Seven studies (five phase III and two phase II) encompassing a total of 2993 patients were included in the analyses. Data from the full-text articles were retrieved for three studies (Checkmate 816, NADIM II, TD-Foreknow), and congress abstracts were used for data extraction for four studies (Neotorch, Keynote 671, Checkmate 77T, AEGEAN) due to availability in three studies and the presence of updated OS data for Keynote 671. Four studies included patients with stage II and III disease, while three included only stage III disease. Nivolumab was used in three studies and the sample size was between 86 and 799. The immunotherapy was used in the neoadjuvant setting only in the Checkmate 816 and TD-Foreknow studies, while in the NADIM II, Keynote 671, Neotorch, Checkmate 77T, and the AEGEAN studies, both neoadjuvant chemoimmunotherapy and adjuvant immunotherapy after surgery was present (Table 1).

### 3.2. The Comparison of EFS and OS with Chemoimmunotherapy versus Chemotherapy

In the pooled analysis of seven studies, the use of neoadjuvant chemoimmunotherapy was associated with a 41% reduction in the risk of progression or death compared to neoadjuvant chemotherapy (HR: 0.59, 95% CI: 0.52–0.66, *p* < 0.0001) (Figure 2). The included studies had a low degree of heterogeneity (I^2^ = 12%), and the sensitivity analyses conducted by the subtraction of the individual studies demonstrated a consistent benefit with immunotherapy.

The data regarding OS were available in four studies (Checkmate 816, Keynote 671, Neotorch, NADIM II). In the pooled analysis of these four studies (total of 1645 patients), the use of neoadjuvant chemoimmunotherapy was associated with a significantly lower risk of death compared to neoadjuvant chemotherapy (HR: 0.67, 95% CI: 0.55–0.82, *p* < 0.0001) (Figure 3). The included studies had a low degree of heterogeneity (I^2^ = 0%), and subtraction of the individual studies demonstrated a consistent benefit with immunotherapy.

### 3.3. Subgroup Analyses for EFS

Subgroup analyses for EFS were conducted according to following parameters: PD-L1 expression (<1 vs. 1–49% vs. >50%), ECOG status (0 vs. 1), age (<65 vs. ≥65 years), stage (stage IB-II vs. stage III), platinum agent (carboplatin vs. cisplatin), histologic type (squamous vs. non-squamous), and smoking status (never smoker vs. current or former smoker). In the subgroup analyses other than PD-L1 expression, the benefit of immunotherapy was consistent across the evaluated without any apparent subgroup difference (*p* > 0.05 for test for subgroup differences) (Figure 4 and Appendix A). However, the benefit of immunotherapy was significantly lower in patients with PD-L1-negative tumors (HR: 0.75, 95% CI: 0.62–0.90, *p* = 0.0020) than with the intermediate level (PD-L1 1–49%) (HR: 0.56, 95% CI: 0.46–0.70, *p* < 0.0001) and high levels (PD-L1 > 50%) (HR: 0.42, 95% CI: 0.33–0.54, *p* < 0.0001) of PD-L1 expression (*p*-value for subgroup difference, *p* = 0.0010) (Figure 4a).

### 3.4. The Comparison of Pathological Complete Response, R0 Resection Rates, and Adverse Events with Chemoimmunotherapy versus Chemotherapy

The data regarding the pCR and R0 resection rates were available in all seven of the included studies. In the pooled analysis of seven studies, the use of chemoimmunotherapy was associated with significantly higher pCR rates compared to chemotherapy (21.8% vs. 3.8%, OR: 7.04, 95% CI: 5.23–9.47, *p* < 0.0001) (Appendix A). The R0 resection rates were also improved with chemoimmunotherapy compared to chemotherapy (OR: 1.63, 95% CI: 1.24–2.14, *p* = 0.0005) (Appendix A). In the pooled data of seven studies, all-grade adverse events were similar in patients treated with chemoimmunotherapy (96.2%) and with chemotherapy (95.8%) (OR: 1.15, 95% CI: 0.80–1.67, *p* = 0.4500) (Figure 5a). However, the high-grade (grade 3 or higher) adverse events were higher with chemoimmunotherapy (45.1%) than with chemotherapy (41.7%) (OR: 1.18, 95% CI: 1.02–1.36, *p* = 0.0300) (Figure 5b). Although there was a trend towards lower pneumonectomy rates in patients treated with chemoimmunotherapy, the difference did not reach statistical significance (OR: 0.76, 95% CI: 0.57–1.00, *p* = 0.0500) (Appendix A). All analyses had a low degree of heterogeneity (I^2^ < 50% for all).

## 4. Discussion

In this meta-analysis of seven randomized clinical trials comparing neoadjuvant chemoimmunotherapy and chemotherapy, we observed that neoadjuvant immunotherapy significantly improves EFS, OS, and pCR rates, with a slight increase in high-grade toxicities. The subgroup analyses demonstrated a consistent benefit with neoadjuvant immunotherapy independent of age, ECOG status, smoking status, and stage, while the benefit of neoadjuvant immunotherapy was lower in patients with PD-L1-negative tumors than patients with moderate or high levels of PD-L1 expression. To our best knowledge, the present meta-analysis is the most up-to-date meta-analysis on the field, and the first meta-analysis incorporating the very recent Checkmate 77T study to the analyses.

The neoadjuvant treatments emerged as a feasible and pragmatic way to apply systemic treatment in patients with localized cancers [23]. The neoadjuvant treatments have significant advantages compared to the adjuvant setting, including the in vivo evaluation of drug sensitivity, improvement in the R0 resections, earlier eradication of the micrometastatic disease, and a higher chance for the completion of systemic treatment [24,25,26,27]. Using ICIs in the neoadjuvant setting could be more beneficial than in the adjuvant setting due to higher tumor neoantigen burden and intact lymphatic drainage before surgery. Furthermore, neoadjuvant treatments aid in the prognostication of the tumors and individualization of the adjuvant treatments, as seen in the low rates of recurrence in patients with HER-2-positive breast cancer and triple-negative breast cancer who had pCR with neoadjuvant treatment [28,29]. However, the uptake of the neoadjuvant treatment was low in NSCLC due to low rates of pCR with neoadjuvant chemotherapy, and the fear of progression during the neoadjuvant setting.

The interest in the neoadjuvant treatment in NSCLC accelerated after the pivotal study with nivolumab by Forde et al. in 2018 [11]. In the study, two doses of nivolumab were associated with major pathological response in 45% and pCR in 15% of the resected tumors. Additionally, nivolumab led to expansion of the T-cell clones both in the tumor and the peripheral blood. Afterward, in the NADIM trial, three cycles of carboplatin plus paclitaxel and nivolumab followed by 1-year adjuvant nivolumab was associated with a 24-month PFS rate of 77.1% [18]. The treatment was not associated with surgery delays and deaths in the cohort (*n* = 46 patients). After these impressive results, later studies were planned with neoadjuvant chemoimmunotherapy. Combining chemotherapy and immunotherapy demonstrated synergism in the treatment of metastatic NSCLC and has a strong biological rationale in clinical practice. Proposed mechanisms for increased efficacy of immunotherapy when combined with chemotherapy are the increased neoantigen presentation, reduction in the immunosuppressive cells, and the activation of the immune effector cells with chemotherapy [30,31,32].

The association with PD-L1 expression and ICI efficacy is context- and treatment-line-dependent in metastatic NSCLC, with studies demonstrating a benefit with ICIs in PD-L1-negative patients in the first-line setting in combination with chemotherapy and in the later lines as monotherapy [33,34]. The data are even more conflicting in the adjuvant setting, with the IMpower 010 study suggesting a higher benefit with higher PD-L1 expression with adjuvant atezolizumab [35], while there was no association with PD-L1 expression with clinical benefit with adjuvant pembrolizumab in the Keynote-091 study [36]. The mounting evidence suggests that PD-L1 is a continuous marker for ICI efficacy and dichotomous cut-offs to predict efficacy are relatively imprecise [37,38]. We observed an EFS benefit with neoadjuvant chemoimmunotherapy compared to chemotherapy in PD-L1 levels across PD-L1 < 1%, PD-L1 1–49%, and PD-L1 > 50%. However, the magnitude of benefit was significantly higher in PD-L1 1–49% and PD-L1 > 50% groups. While this result still supports the use of neoadjuvant chemoimmunotherapy in patients with PD-L1-negative tumors, novel approaches are needed to further improve the prognosis for these patients, as well as a need for further studies to delineate the predictive role of PD-L1 expression for neoadjuvant chemoimmunotherapy.

There are several research gaps in the neoadjuvant chemoimmunotherapy field in NSCLC. First, the optimal duration of adjuvant immunotherapy after neoadjuvant treatment is yet unknown. The available studies significantly differed in this regard. While only three cycles of neoadjuvant immunotherapy with no adjuvant treatment in the Checkmate 816 study could be a more cost-effective and feasible approach, there was an obvious need to improve the prognosis of patients without a pCR with neoadjuvant treatment across most studies [15,17,18]. Clinical trials permitting individualized treatment duration after surgery are urgently needed to clarify this issue. The incorporation of the ctDNA to optimize adjuvant treatment after neoadjuvant chemoimmunotherapy could be potentially useful, although prospective evidence is lacking. The available studies had relatively short follow-up times, and mature overall survival data are needed to completely define the benefit of neoadjuvant chemoimmunotherapy, especially the adjuvant part of the treatment. Another important point is the possibility of using dual immunotherapy with chemotherapy to improve the outcomes further. The combination of nivolumab plus ipilimumab had higher MPR rates (50% vs. 24%) and pCR rates (38% vs. 10%) compared to nivolumab in the phase II Neostar trial [39]. The combination of nivolumab plus ipilimumab with chemoradiotherapy was associated with a pCR rate of 63% in surgery in the recent phase II INCREASE trial, including patients with resectable and borderline resectable NSCLC [40]. Considering the higher benefit of combination in the Neostar trial and the feasibility of using nivolumab plus ipilimumab in combination with chemotherapy as in the Checkmate 9LA study in the metastatic stage [41], further research with combination immunotherapy and chemotherapy is expected. Another important potential will be potential for surgery in patients with conventionally unresectable disease. Over 30% of the patients in the Neotorch study [20], and over 15% of the patients in the Checkmate 77T study, had multistation N2 disease [21]. Separate subgroup analyses from Checkmate 77T for the patients with multistation N2 disease demonstrated a significant benefit in these patients (HR: 0.43, 95% CI: 0.21–0.88). These data, if supported by further studies (ongoing NEOSUN trial, NCT04943029), could lead to use of surgery for multistation N2 disease. Lastly, the selection of adjuvant vs. neoadjuvant immunotherapy is a critical question. The recent IMPower 010 and KEYNOTE-091 studies demonstrated a DFS benefit with the use of one-year adjuvant treatment [35,36]. Although the studies differed regarding the biomarker results, we have phase III evidence for the use of adjuvant immunotherapy in resected NSCLC. While the pharma interest in a study comparing neoadjuvant vs. adjuvant immunotherapy would be low, independently funded research is needed to define the best use of ICIs in localized NSCLC.

## 5. Conclusions

In conclusion, the available evidence suggests a statistically significant and clinically meaningful EFS benefit and possibly an OS benefit with neoadjuvant chemoimmunotherapy with a slight increase in high-grade toxicities. Further research is needed to define the optimal duration of immunotherapy after surgery and the overall survival benefit of neoadjuvant immunotherapy with longer follow-ups.

## Figures and Tables

**Figure 1 cancers-16-00156-f001:**
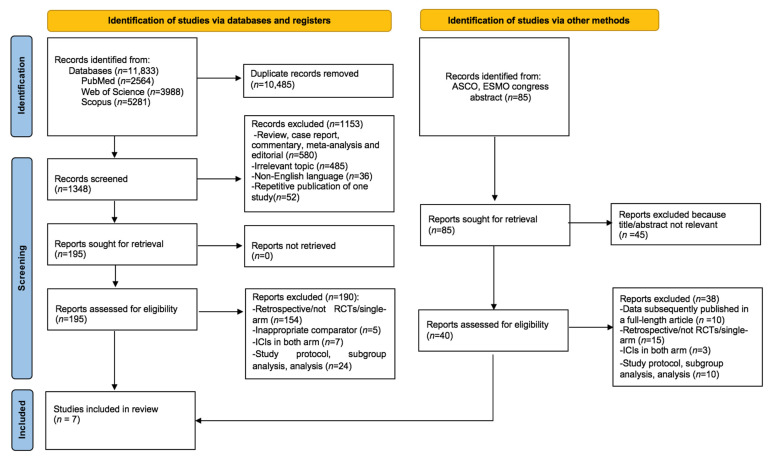
PRISMA flow chart showing the selection of articles for systematic review and meta-analysis.

**Figure 2 cancers-16-00156-f002:**
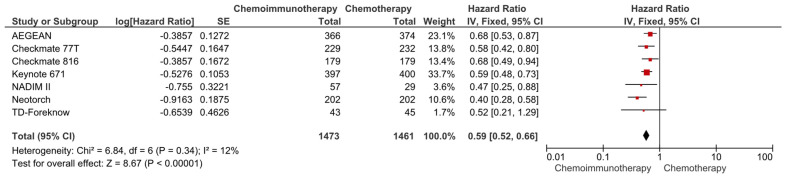
Forest plots for EFS between neoadjuvant chemoimmunotherapy and chemotherapy regimens in resected NSCLC patients.

**Figure 3 cancers-16-00156-f003:**
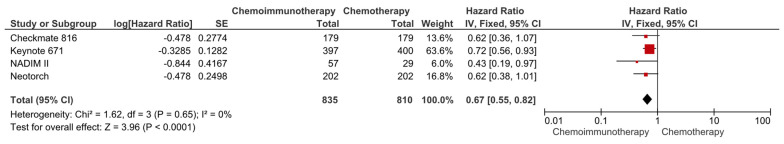
Forest plots for OS between neoadjuvant chemoimmunotherapy and chemotherapy regimens in resected NSCLC patients.

**Figure 4 cancers-16-00156-f004:**
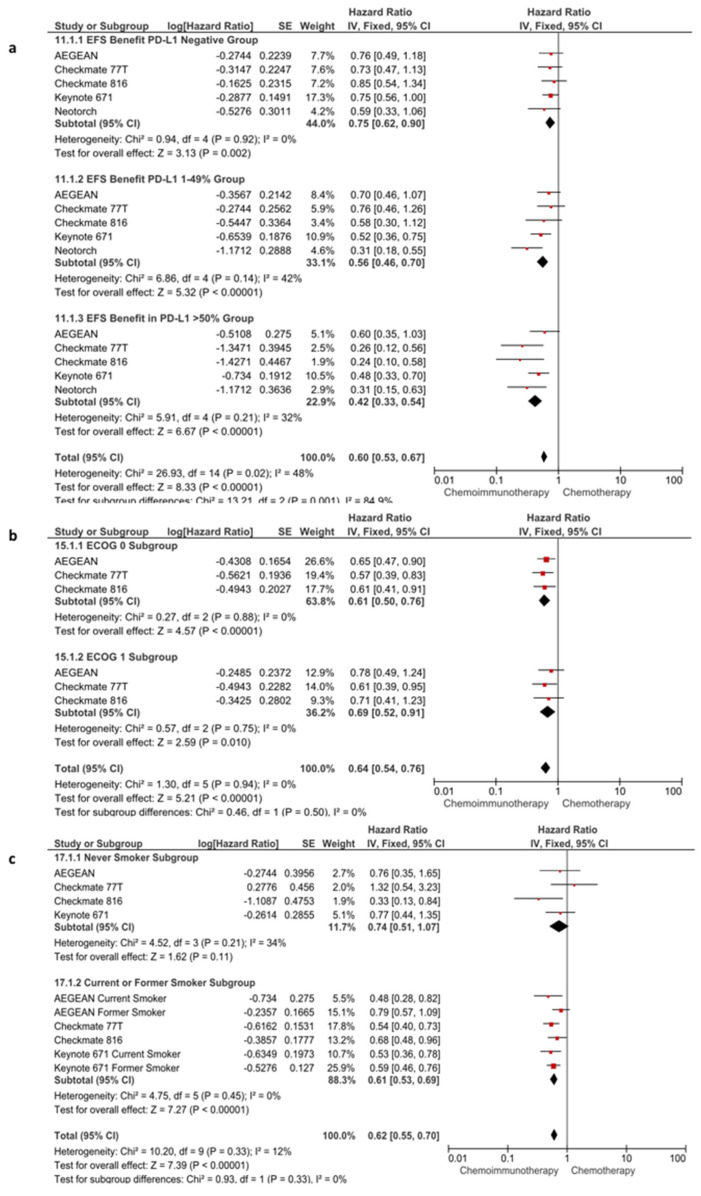
Subgroup analyses according to PD-L1 expression (**a**), ECOG status (**b**) and smoking status (**c**) in EFS.

**Figure 5 cancers-16-00156-f005:**
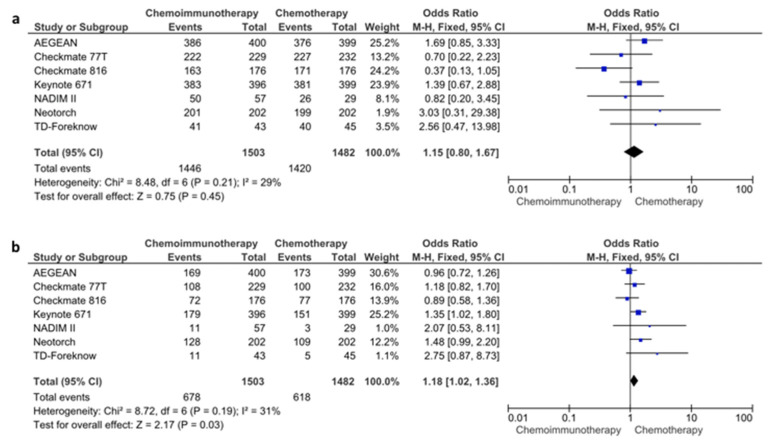
All-grade (**a**) and high-grade (**b**) adverse event rates for chemoimmunotherapy and chemotherapy arms.

**Table 1 cancers-16-00156-t001:** Characteristics of included studies.

Trial Name, Year	Phase	Clinical Stage	Total Number of Patients	Study Design	Experimental Arm (*n*)	Control Arm (*n*)	Surgical Resection Rate	R0 Resection	pCR, No./Total No. (%)	OS (Median or %)	EFS/DFS/PFS (Median or %	Grade ≥ 3 TRAEs, (%)	Fatal AEs, No.	Median Follow-Up (Months)	Additional Comments
CheckMate 816, 2022 [15]	III	IB-IIIA	358	Neoadjuvant only	Nivolumab + Chemotherapy (179)	Chemotherapy (179) (NSQ: pemetrexed + cisplatin/paclitaxel + carboplatin, SQ: gemcitabine + sisplatin/paclitaxel + carboplatin)	83.2% vs. 75.4%	83.2% vs. 77.8%	43/179 (24) vs. 4/179 (2.2)	NR	31.6 vs. 20.8 months (median)	33.5 vs. 36.9	0 vs. 3	29.5	No additional ICI in the adjuvant setting
NADIM II, 2023 [18]	II	IIIA-IIIB	86	Neoadjuvant plus adjuvant	Nivolumab + Chemotherapy (57)	Chemotherapy (29) (paclitaxel + carboplatin)	96.2% vs. 100%	94.3% vs. 85.0%	21/57(37) vs. 2/29 (7)	1 year (98.2 vs. 82.1)	1 year PFS (89.5 vs. 58.6)	19 vs. 10	1 vs. 0	26.1	The ICIs were continued for six months for the adjuvant setting in the experimental arm
KEYNOTE-671, 2023 [19]	III	II-IIIB	797	Neoadjuvant plus adjuvant	Pembrolizumab + Chemotherapy (397)	Chemotherapy (400) (NSQ: cisplatin + pemetrexed, SQ: cisplatin + gemcitabine)	98.5% vs. 95.3%	92% vs. 84.2%	72/397 (18.1) vs. 16/400 (4)	3 year (71.3 vs. 64.0) 4 years (67.1 vs. 51.5)	3 year EFS (54.3 vs. 35.4) 4 year EFS (48.4 vs. 26.2)	45.2 vs. 37.8	1 vs. 0	36.6	All patients treated with cisplatin-based combinations, requiring cisplatin eligibility for trial enrollment
TD-FOREKNOW, 2023 [17]	II	IIIA-IIIB	88	Neoadjuvant only	Camrelizumab + Chemotherapy (43)	Chemotherapy (45) (nab-paclitaxel plus cisplatin, carboplatin or nedaplatin)	93% vs. 93.3%	92.5% vs. 85.7%	14/43 (32.6) vs. 4/45 (8.9)	NR	1 year EFS (93.0 vs. 76.9) 2 year EFS (76.9 vs. 67.6)	25.6 vs. 11.1	0 vs. 0	14.1	The primary endpoint was the pCR rate. The EFS was a secondary endpoint. No treatment in the adjuvant setting
AEGEAN, 2023 [16]	III	II-IIIB	799	Neoadjuvant plus adjuvant	Durvalumab + Chemotherapy (400)	Chemotherapy (399) (NSQ: pemetrexed + cisplatin or carboplatin, SQ: carboplatin + paclitaxel/gemcitabine + cisplatin or carboplatin)	80.6% vs. 80.7%	94.7% vs. 91.3%	63/366 (17.2) vs. 16/374 (4.3)	NR	1 year EFS (73.4 vs. 64.5)	32.3 vs. 33.1	7 vs. 2	11.7 m0	-
Neotorch, 2023 [20]	III	II-III	404	Neoadjuvant plus adjuvant	Toripalimab + Chemotherapy (202)	Chemotherapy (202)	82.2% vs. 73.3%	95.8% vs. 92.6%	50/202 (24.8) vs. 2/202 (1)	1 year (94.4 vs. 89.6)	1 year EFS (84.4 vs. 57.0)	63.4 vs. 54.0	0 vs. 2	18.25 mo	Included patients with multi-station N2 disease, 33 and 31% of the patients in the experimental and control arms had multi-station N2 disease, respectively
CheckMate 77T, 2023 [21]	III	IIA-IIIB	461	Neoadjuvant plus adjuvant	Nivolumab + Chemotherapy (229)	Chemotherapy (232) (NSQ: pemetrexed + cisplatin or carboplatin/carboplatin + paclitaxel, SQ: cisplatin + docetaxel/ carboplatin + paclitaxel)	78% vs. 77%	89% vs. 90%	58/229 (25.3) vs. 11/232 (4.7)	NR	1 year EFS (73 vs. 59)	27.0 vs. 23.0	2 vs. 0	25.4	Included patients with multi-station N2 disease and had separate subgroup data for these patients

## Data Availability

Data were extracted from the original studies included in this systematic review. All data used for statistical analyses are presented in the manuscript and its Appendix A.

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
