# Peer review of "The Efficacy and Safety of Neoadjuvant Immunotherapy in Patients with Non-Small Cell Lung Cancer"

_cancers, 2023, doi:10.3390/cancers16010156_

Round 1

Reviewer 1 Report

Comments and Suggestions for Authors

In their manuscript „The Efficacy and Safety of Neoadjuvant Immunotherapy in Patients with Non-Small Cell Lung Cancer“ authors performed systemic review and meta-analysis with the aim to investigate the overall benefit of neoadjuvant chemoimmunotherapy in patients with NSCLC. The authors reported that neoadjuvant chemoimmunotherapy improved clinical outcome of patients (EFS and OS), however is associated with high-grade adverse events, compared to neodjuvant chemotherapy.

The manuscript is written in a clear and understandable way. The topic of the manuscript is of immense importance and I believe will contribute greatly to the field.

I have only few minor comments. In the Table 1, the authors should adjust the font size to fit the table margins and add the relevant references that relate to corresponding clinical trials. There are also few typing errors throughout the text, so I recommend the authors to carefully read the paper and revise it before publishing.

Comments on the Quality of English Language

As mentioned, there are also few typing errors throughout the text, so I recommend the authors to carefully read the paper and revise it before publishing.

Author Response

Many thanks to the Editor and reviewers for their constructive criticism and insightful comments. We diligently worked to constructively address each of these comments. This manuscript has been read and approved by all the authors. Our responses to the reviewer’s comments are given point by point:

Comments from the Editors and Reviewers:

General Comments of Reviewer 1. “In their manuscript “The Efficacy and Safety of Neoadjuvant Immunotherapy in Patients with Non-Small Cell Lung Cancer“ authors performed systemic review andmeta-analysis with the aim to investigate the overall benefit of neoadjuvant chemoimmunotherapy in patients with NSCLC. The authors reported that neoadjuvant chemoimmunotherapy improved clinical outcome of patients (EFS and OS), however is associated with high-grade adverse events, compared to neodjuvant chemotherapy. The manuscript is written in a clear and understandable way. The topic of the manuscript is of immense importance and I believe will contribute greatly to the field.

Response: We thank the reviewer for the overall very favorable review of our manuscript and the constructive comments. The incorporation of the comments and corresponding revisions have enhanced the quality of our manuscript.

Comment #1 of Reviewer 1. “I have only few minor comments. In the Table 1, the authors should adjust the font size to fit the table margins and add the relevant references that relate to corresponding clinical trials.”

Response : Thank you very much for your comment. We have adjusted the font size in Table 1 to fit within the margins. We have also added the relevant references for the clinical trials mentioned in Table 1.

Comment #2 of Reviewer 1.  There are also few typing errors throughout the text, so I recommend the authors to carefully read the paper and revise it before publishing.

Response: We have meticulously read through the entire manuscript to identify and correct the typographical errors.

Reviewer 2 Report

Comments and Suggestions for Authors I found this meta-analysis to be a useful exercise in its attempt to strengthen results of the 6 analyzed studies. I have no concerns about the data collection or the statistical processes used in this analysis/study. In terms of the discussion and conclusions, I would recommend adding something about the impact of this meta-analysis on strengthening or weakening the conclusions reached in the individual studies included in the analysis. I'm not so sure the suggestions put forward in the conclusions section (need for further research...,longer follow-up...) can actually be deduced from this analysis. Comments on the Quality of English Language PS: I found a very few grammatical and/or spelling mistakes.

Author Response

Many thanks to the Editor and reviewers for their constructive criticism and insightful comments. We diligently worked to constructively address each of these comments. This manuscript has been read and approved by all the authors. Our responses to the reviewer’s comments are given point by point:

Comments from the Editors and Reviewers:

General Comments of Reviewer 2.  “I found this meta-analysis to be a useful exercise in its attempt to strengthen results of the 6 analyzed studies. I have no concerns about the data collection or the statistical processes used in this analysis/study. In terms of the discussion and conclusions, I would recommend adding something about the impact of this meta-analysis on strengthening or weakening the conclusions reached in the individual studies included in the analysis. I'm not so sure the suggestions put forward in the conclusions section (need for further research...,longer follow-up...) can actually be deduced from this analysis.

Response: We highly appreciate the reviewer’s constructive comments. We updated our search to include the latest literature, incorporating the very recent Checkmate 77T study for the most comprehensive analysis to date. We have added a section to the discussion that specifically addresses the impact of our meta-analysis on the strength of the conclusions drawn in the individual studies. We have meticulously read through the entire manuscript to identify and correct the typographical errors.

Reviewer 3 Report

Comments and Suggestions for Authors

The authors that review and summary the immunotherapy and compared with neoadjuvant chemoimmunotherapy in non-small cell lung cancer (NSCLC) for 2532 patients. They conducted a systematic review and meta-analysis to evaluate the benefit of adding immunotherapy to neoadjuvant chemotherapy in localized NSCLC patients. They performed the meta-analyses with the generic inverse-variance method with a fixed effects model. They concluded that with neoadjuvant chemoimmunotherapy with a slight increase in high-grade toxicities and that is needed to define optimal duration of immunotherapy after surgery, and the overall survival benefit of neoadjuvant immunotherapy with longer follow-up. Some comment as following:

1.     In the Table 1 should be modified for more ease to check the points.

2.     In the Discussion should provide the new concept and some suggested on the neoadjuvant chemoimmunotherapy.

3.      In those treatment, may be mention the immune response, if it is possible.

Comments on the Quality of English Language

Minor editing of English language required.

Author Response

Many thanks to the Editor and reviewers for their constructive criticism and insightful comments. We diligently worked to constructively address each of these comments. This manuscript has been read and approved by all the authors. Our responses to the reviewer’s comments are given point by point:

Comments from the Editors and Reviewers:

General Comments of Reviewer 3.  The authors that review and summary the immunotherapy and compared with neoadjuvant chemoimmunotherapy in non-small cell lung cancer (NSCLC) for 2532 patients. They conducted a systematic review and meta-analysis to evaluate the benefit of adding immunotherapy to neoadjuvant chemotherapy in localized NSCLC patients. They performed the meta-analyses with the generic inverse-variance method with a fixed effects model. They concluded that with neoadjuvant chemoimmunotherapy with a slight increase in high-grade toxicities and that is needed to define optimal duration of immunotherapy after surgery, and the overall survival benefit of neoadjuvant immunotherapy with longer follow-up. Some comment as following:

Response : We are grateful for your thorough review and valuable suggestions, which have guided significant improvements in our manuscript

Comment #1 of Reviewer 3.  In the Table 1 should be modified for more ease to check the points.

Response: We have revised Table 1 to enhance its clarity and facilitate an easier review of the key points.

Comment #2 of Reviewer 3.  In the Discussion should provide the new concept and some suggested on the neoadjuvant chemoimmunotherapy.

Response: In the Discussion section, we have introduced a new concepts arising from our findings, particularly in the context of neoadjuvant chemoimmunotherapy. We explore how these concepts may impact current understanding and practice. Additionally, we have included a paragraph that offers suggestions for the the incorporation of the ctDNA to optimize adjuvant treatment after neoadjvuvant chemoimmunotherapy, which we hope will stimulate further investigation in this promising area (Line 618-623).

Comment #3 of Reviewer 3.  In those treatment, may be mention the immune response, if it is possible.

Response: We recognize the importance of discussing the immune response in the context of neoadjuvant chemoimmunotherapy. To address this, we have included a discussion on the potential immune responses elicited by these treatments. (Line 575-578)

Reviewer 4 Report

Comments and Suggestions for Authors

In this interesting review authors reviewed the efficacy and safety of neoadjuvant immunotherapy in NSCLC patients. I only have 1 concern:

1   1)  Authors should include more information in the introduction

Author Response

Many thanks to the Editor and reviewers for their constructive criticism and insightful comments. We diligently worked to constructively address each of these comments. This manuscript has been read and approved by all the authors. Our responses to the reviewer’s comments are given point by point:

Comments from the Editors and Reviewers:

General Comments of Reviewer 4.  In this interesting review authors reviewed the efficacy and safety of neoadjuvant immunotherapy in NSCLC patients. I only have 1 concern: Authors should include more information in the introduction.

Response: Thank you for your kind words regarding our review, and for your valuable suggestion to enhance the introduction of our manuscript. In response to your comment, we have expanded the introduction to provide a more comprehensive background on the subject (Line 89-95).

Round 2

Reviewer 4 Report

Comments and Suggestions for Authors

Authors addressed all my concerns.